# Analysis of Lake Stratification and Mixing and Its Influencing Factors over High Elevation Large and Small Lakes on the Tibetan Plateau

Binbin Wang [1,2,*], Yaoming Ma [1,2,3,4,5,6,*], Yan Wang [7], Lazhu [8], Lu Wang [9], Weiqiang Ma [1,3,4] and Bob Su [10]

1   State Key Laboratory of Tibetan Plateau Earth System, Resources and Environment (TPESRE), Institute of Tibetan Plateau Research, Chinese Academy of Sciences, Beijing 100101, China; wqma@itpcas.ac.cn
2   National Observation and Research Station for Qomolongma Special Atmospheric Processes and Environmental Changes, Dingri 858200, China
3   College of Earth and Planetary Sciences, University of Chinese Academy of Sciences, Beijing 100049, China
4   College of Atmospheric Science, Lanzhou University, Lanzhou 730000, China
5   Kathmandu Center of Research and Education, Chinese Academy of Sciences, Beijing 100101, China
6   China-Pakistan Joint Research Center on Earth Sciences, Chinese Academy of Sciences, Islamabad 45320, Pakistan
7   Key Laboratory of Land Surface Pattern and Simulation, Institute of Geographic Science and Natural Resources Research, Chinese Academy of Sciences, Beijing 100101, China
8   Research Center for Ecology, College of Science, Tibet University, Lhasa 850000, China
9   Hebei Province Tangshan Caofeidian Meteorological Bureau, Tangshan 063000, China
10  Faculty of Geo-Information Science and Earth Observation, University of Twente, 7500 AE Enschede, The Netherlands
*   Correspondence: wangbinbin@itpcas.ac.cn (B.W.); ymma@itpcas.ac.cn (Y.M.)

**Abstract:** Lake stratification and mixing processes can influence gas and energy transport in the water column and water–atmosphere interactions, thus impacting limnology and local climate. Featuring the largest high-elevation inland lake zone in the world, comprehensive and comparative studies on the evolution of lake stratification and mixing and their driving forces are still quite limited. Here, using valuable temperature chain measurements in four large lakes (Nam Co, Dagze Co, Bangong Co, and Paiku Co) and a "small lake" adjacent to Nam Co, our objectives are to investigate the seasonal and diurnal variations of epilimnion depth ($E_p$, the most important layer in stratification and mixing process) and to analyze the driving force differences between "small lake" and Nam Co. Results indicate that $E_p$ estimated by the methods of the absolute density difference (<0.1 kg m$^{-3}$) from the surface and the Lake-Analyzer were quite similar, with the former being more reliable and widely applicable. The stratification and mixing in the four large lakes showed a dimictic pattern, with obvious spring and autumn turnovers. Additionally, the stratification form during heat storage periods, with $E_p$ quickly locating at depths of approximately 10–15 m, and, after that, increasing gradually to the lake bottom. Additionally, the diurnal variation in $E_p$ can be evidenced both in the large and small lakes when temperature measurements above 3 m depth are included. For Nam Co, the dominant influencing factors for the seasonal variation of $E_p$ were the heat budget components (turbulent heat fluxes and radiation components), while wind speed only had a relatively weak positive correlation (r = 0.23). In the "small lake", radiation components and wind speed show high negative (r = −0.43 to −0.59) and positive (r = 0.46) correlation, with rare correlations for turbulent heat flux. These reported characteristics have significance for lake process modeling and evaluation in these high-elevation lakes.

**Keywords:** lake stratification; epilimnion depth; large and small lake; high-elevation lakes; Tibetan Plateau

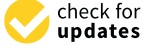

## 1. Introduction

Lake stratification and mixing is one of the most basic and fundamental processes in limnology and hydrometeorology. Generally, the water column of lakes during the stratification period can be classified into three layers: epilimnion, metalimnion, and hypolimnion. The epilimnion (also known as the mixing layer with a homogeneous distribution of temperature) is influenced directly by thermally induced convective cooling and wind-induced mechanical dynamics [1,2]. The metalimnion (also known as the thermocline, where temperature decreases rapidly with depth) could retard the gas, energy, and material transfers between the upper epilimnion and the bottom hypolimnion. The hypolimnion is the deep-water layer where hypoxia or anoxia mainly appears [3], and such extreme conditions can be alleviated through lake mixing events. Recent studies have shown that lake mixing can channel the epilimnion and hypolimnion and release large amounts of potent greenhouse gases ($CO_2$ and $CH_4$) into the upper surface layer and the atmosphere, making lakes weak net carbon sources [4,5]. Thus, lake stratification and mixing have been found to determine the algal distribution, the photosynthesis rates, and the establishment of a food web basis [1,3,6,7], and it can also impact the lake-atmosphere energy and water flux exchange and consequently the regional climate [8].

The seasonal and diurnal evolution of lake stratification and mixing is mainly determined by the balance between the stabilizing effects of surface heating and the destabilizing effects of surface cooling and turbulent mixing generated by the wind, with the former suppressing mixing and the latter enhancing mixing [8,9]. Thus, positive net heat inputs during the day generally can heat the water column and form a warming and stabilized surface layer and a positive buoyancy flux, while net heat loss at night can cool the surface waters and result in vertical mixing because of temperature-induced density instability. Observations indicate that surface cooling can lead to full convective mixing of the water column each night in a shallower lake [9], while stronger winds have been reported to deepen the epilimnion, shorten the stratified season, and warm the deep hypolimnion [10]. Thus, in situ measurements have reported that the diurnal oscillation of water temperature can be influenced by continuous heat fluxes and occasional forced diurnal variation in water temperature at depths from approximately 7 m to 20 m can also be influenced by noncontinuous periodic winds [11].

Solar radiation and wind speed are the most direct influencing factors for the development of lake mixing and stratification, where heat-induced buoyancy and wind-induced internal waves can promote vertical temperature variations. In addition to the meteorological factors, lake morphometry (lake depth, lake fetch, and lake topography) and water color parameters (water clarity, light attenuation, and dissolved organic matter) can also impact lake stratification and mixing [12–16]. For example, the macrophyte canopy can influence light attenuation and, in turn, impacts the epilimnion depth [9]. The summer epilimnion depths are related to lake size and water clarity over 21 Canadian Shield lakes, where surface area primarily determines and transparency significantly modifies the relationship [17]. Moreover, moderate and heavy rainfall events can reduce the surface water temperature and generate increased inflows, thus reducing the thermal stability of the water column and deepening the epilimnion depth [12]. From 6 July to 18 November 2018, the epilimnion depth of a small and deep lake (0.82 km$^2$ and maximum depth of 46 m) showed a deepening variation from approximately 10 m to 16 m [11], while the epilimnion depth showed a gradual decrease from April to June and a gradual increase from August to November in two European lakes [1].

Featured as the "Third Pole" of Earth and "Asian water tower", the TP hosts the world's largest high-elevation inland lake zone, which amounts to more than 50% of the lake area in China [18]. Influenced by the high elevation and low air temperature, the majority of lakes on the TP are dimictic and generally covered in ice during winter [19]. However, among these high-elevation lakes, only lake temperature chains have been observed on a few large lakes, including Bangong Co, Dagze Co [20,21], Paiku Co [22], Nam Co [23,24], Ngoring Lake [25], and Qinghai Lake [26]. The analysis of lake stratification and mixing

is quite limited. Wang et al. [23] divided the stratification and mixing conditions of Nam Co into six phases, analyzed the characteristics of lake states ($T$ vs. $T_{md}$; ice cover; heat content; stability) under each phase, and highlighted the significance of radiation-driven convection in the stratification and mixing variations. Wang et al. [21] analyzed the seasonal variation in lake stability variables, including Schmidt stability, Wedderburn number, and lake number, and concluded no obvious changes in the stratification and mixing regimes by 3–4 years of in situ measurements in Bangong Co and Daze Co.

The atmospheric forcing over lakes on the TP has strong diurnal variations, with obvious surface warming during the day and convective cooling at night; thus, it is necessary to investigate whether diurnal variation in epilimnion depth could be evidenced by in situ temperature chains measurements. What are the general characteristics of the dynamic evolution of stratification and mixing in these high-elevation lakes? Are there differences in the driving forces of the stratification and mixing conditions between small and large lakes? To answer the above questions, the in situ measurements of lake temperature chains and surrounding meteorological variables were collected over four large lakes (Nam Co, Paiku Co, Bangong Co, and Dagze Co) and a "small lake" adjacent to Nam Co ("small lake" for short afterward). Additionally, our objectives are to understand the evolution of stratification and mixing processes in these high-elevation lakes and to figure out the differences in dominating driving forces to the variation of epilimnion depth between small lakes and large lakes. The configuration of water temperature chains and meteorological variables, stratification and mixing parameters, and related influencing factors are briefly introduced in Section 2. The comparative study on lakes' thermal stratification and mixing evolution in each lake is introduced in Section 3.1. The comparative study on the seasonal and diurnal variations of stratification and mixing variations (epilimnion depth and lake stability parameters) are shown in Section 3.2. The dominant contributing factors for stratification and mixing variations in Nam Co and the "small lake" are analyzed in Section 3.3. The discussion and conclusions are listed in Sections 4 and 5, respectively.

## 2. Materials and Methods

### 2.1. Study Sites and Measurement Descriptions

The TP hosts a total lake number of more than 1400 lakes larger than 1 km$^2$, where approximately 100 lakes are larger than 100 km$^2$ [27]. Accompanied by the growing interest in high-elevation lake thermal processes, water temperature chain measurements have been deployed more frequently to study the physical and chemical properties and regional climate effects of lakes, e.g., in Bangong Co, Dagze Co [20], Nam Co [24,28], and Paiku Co [22]. In this study, we collected temperature chain measurements in Bangong Co (July 2012 to August 2013, hourly), Dagze Co (August 2012 to August 2013, hourly), Nam Co (November 2011 to June 2014, daily), and Paiku Co (June 2016 to May 2017, hourly) for a comparative study of their seasonal variations in stratification and mixing processes. Moreover, high vertical resolution temperature chain measurements in Nam Co (10 layers, July to November 2015 and 2016, hourly [24]) and a "small lake" adjacent to Nam Co (10 layers, May to November 2021, hourly) were also included, with measuring water depths of 35 m and 14 m, respectively.

The multiyear average monthly meteorological variables can be obtained via long-term measurements at comprehensive observation stations [29] close to the studied high-elevation lakes, including (1) the Ngari Desert Observation and Research Station, CAS (NADORS), (2) Shuanghu station, (3) the Nam Co Monitoring and Research Station for Multisphere Interactions, CAS (NAMORS), and (4) the Qomolangma Atmospheric and Environmental Observation and Research Station, CAS (QOMS). The meteorological variables are collected, and the abnormal measurements outside the normal limits have been removed. Only data with good quality has been used. Additional meteorological measurements on a small island in Nam Co are also available [24]. The instrument configurations can resort to [24,29]. The locations and shapes of the five lakes and the positions of the meteorological stations can be found in Figure 1. The attributes and measurement settings

of the case study lakes can be found in Table 1, and the descriptions of lake environments are as follows.

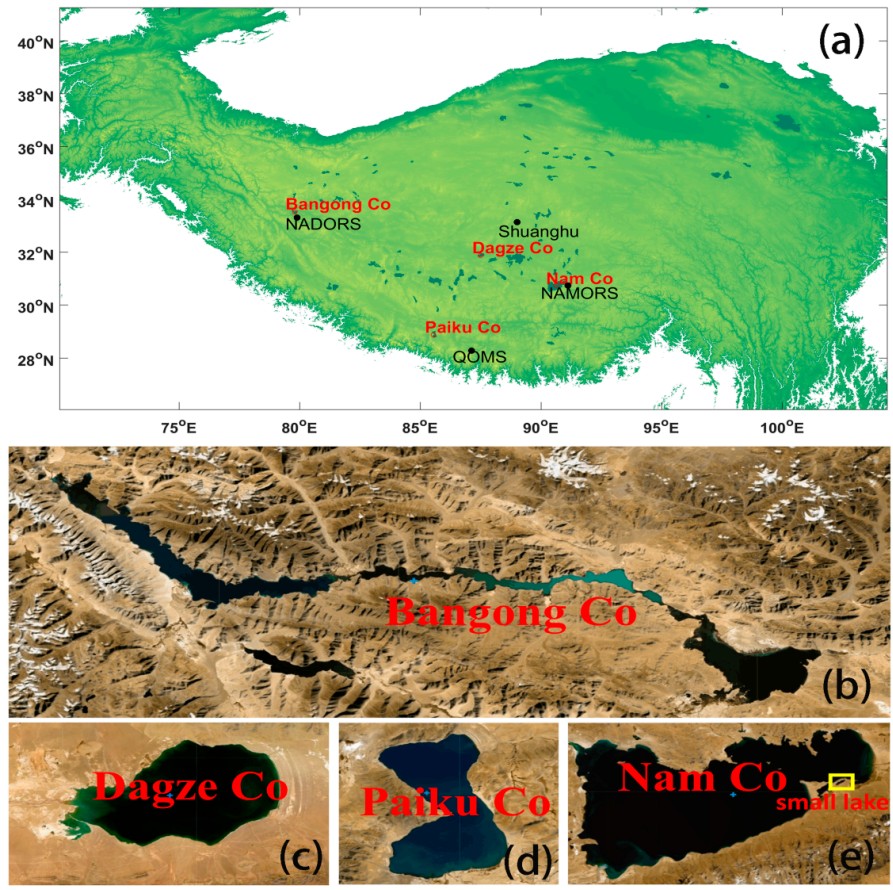

**Figure 1.** The locations of the case study lakes (red circles) and comprehensive observation stations (black dots) on the TP (**a**); the shapes of Bangong Co (**b**), Dagze Co (**c**), Paiku Co (**d**), and Nam Co (**e**); the "small lake" adjacent to Nam Co is marked in the yellow rectangular.

**Table 1.** The attribute and measurement settings of the case study lakes.

| Lakes | Positions | Area (km²); Max Depth (m); Elevation (m) | Observation Periods; Temporal Resolutions | Observation Depths (m) | Sensors |
|---|---|---|---|---|---|
| Nam Co | 33°42′ N 90°39′ E | 2020; 100; 4715; | (1) November 2011 to June 2014; Daily; (2) August 2015 to November 2015; Hourly; July 2016 to November 2016; Hourly; | (1) 3, 6, 16, 21, 26, 31, 36, 56, 66, 83; (2) 0.5, 1.5, 3, 6, 10, 15, 20, 25, 30, 35; | (1) VEMCO water temperature loggers (Minilog-II-T) (2) HOBO water temperature loggers (U22-001) |
| Paiku Co | 28°54′ N 85°35′ E | 280; 72.8; 4590; | June 2016 to May 2017; Hourly; | 0.4, 5, 10, 15, 20, 30, 40; | HOBO water temperature loggers (U22-001) |
| Bangong Co | 33°12′ N 79°12′ E | 627; 41.7; 4220. | July 2012 to August 2013; Hourly; | 5, 6, 7, 8, 9, 10, 11, 13, 15, 17, 19, 21, 23, 26, 29, 32, 35; | HOBO water temperature loggers (U22-001) |
| Dagze Co | 31°54′ N 87°32′ E | 245; 38; 4450; | August 2012 to August 2013; Hourly; | 4, 5, 6, 7, 8, 9, 10, 11, 13, 15, 17, 19, 21, 23, 26, 29, 32, 35; | HOBO water temperature loggers (U22-001) |
| "small lake" | 30°47′ N 90°58′ E | 1.4; 14; 4715; | May 2021 to November 2021; Hourly; | 0.5, 1, 2, 3, 4, 6, 8, 10, 12, 14; | HOBO water temperature loggers (MX2201) |

Bangong Co (33°12′ N, 79°12′ E; 4220 m a.s.l) is a tectonic lake located in the western part of the TP, which is mainly dominated by westerlies and has a total precipitation amount of approximately 90 mm [20]. It has a maximum depth of 41.7 m and an area of 627 km$^2$. The lake has a long stripe shape, which stretches from the southeast to northwest and approaches approximately 150 km in length. The salinity is only 0.47 g L$^{-1}$ and has a measured Secchi-depth value of approximately 14 m during field experiment. NADORS (33°23′ N, 79°42′ E, 4270 m a.s.l.) located 10 km south of Bangong Co and has been established since 2010 [29]. The multiyear annual average air pressure, wind speed, and air temperature are 606 hPa, 2.6 m s$^{-1}$, and 1.8 °C, respectively. The wind speed is much higher from February to May and relatively lower from October to January.

Dagce Co (31°54′ N, 87°32′ E; 4450 m a.s.l.) is located in the central part of the TP and has a surface area of 245 km$^2$ and a maximum depth of 38 m. It is a brackish lake with a salinity value of approximately 16 g L$^{-1}$ from the lake surface to approximately 25 m, a layer with rapidly increasing salinity from 16 g L$^{-1}$ at approximately 25 m to 21.4 g L$^{-1}$ at approximately 29 m, and a layer with a nearly constant value of 21.4 g L$^{-1}$ from 30 m to the lake bottom [20]. The lake is fed mainly by precipitation and the Bogcarg Zangbo River, with estimated epilimnion depths of 16 m to 23 m and an estimated Secchi depth value of 6 m for one day field experiment. Shuanghu station (33°13′ N, 88°49′ E, 4993 m a.s.l.) is approximately 180 km away in the northeast direction of the lake, and the estimated multiyear averaged air pressure, wind speed, and air temperature are 553 hPa, 4.9 m s$^{-1}$, and −4.9 °C, respectively.

Paiku Co (28°54′ N, 85°35′ E, 4590 m a.s.l.) is located in the southern part of the TP and has an area of 280 km$^2$ and a maximum depth of 72.8 m. It is a brackish lake with a salinity value of 1.7 g L$^{-1}$. The lake is influenced by westerly and summer monsoons (May to October) and is fed by precipitation and glacier-melted water. QOMS station (28°22′ N, 86°57′ E, 4276 m a.s.l.) is 140 km away in the southeast direction of the lake and has long-lasting measurements since 2005. The multiyear averaged air pressure, wind speed, and air temperature are 605 hPa, 2.9 m s$^{-1}$, and 4.1 °C, respectively.

Nam Co (33°42′ N, 90°39′ E, 4715 m a.s.l.) is the second largest lake in the central TP and is located to the north of Nyainqentanglha Mountain. The lake has an area of approximately 2020 km$^2$ and a maximum depth of nearly 100 m [24]. It is a brackish lake with a salinity value of 1.5 g L$^{-1}$, and a Secchi-depth value of approximately 6 m. The annual precipitation (approximately 400 mm) and the inflow supplied from glacier-melted water during the warm seasons are the water input, while evaporation is the water loss item in this inner flow lake [24]. NAMORS (30°46′ N, 90°90′ E, 4730 m a.s.l.) locates 1.5 km away to the southeast bank of the lake and has been established since 2006. The multiyear averaged air pressure, wind speed, and air temperature are 571 hPa, 3.4 m s$^{-1}$, and −0.5 °C, respectively.

A "small lake" adjacent to Nam Co is just 500 m away in the north direction of NAMORS, with an area of approximately 1.4 km$^2$ and a maximum depth of 14 m [24,30]. The ice-covered season stretches from the middle of November to the beginning of April and has an average evaporation value of 812 mm during the open water season [31].

The multiyear averaged meteorological variables at the four stations are plotted in Figure 2. Generally, air temperature ($T_a$) and air pressure ($P$) correspond to the elevation of the stations (Figure 2a,b), with the lowest $T_a$ and $P$ at Shuanghu station (approximately 5000 m a.s.l.) and highest at the NADORS or QOMS stations (approximately 4250 m a.s.l.). NADORS is mainly influenced by westerlies, with a much higher wind speed during February to May and a relatively calm winter (Figure 2c), and it is quite dry, with an annual averaged relative humidity value of 29% (Figure 2d). For QOMS, Shuanghu, and NAMORS, the South Asia monsoon plays an important role, and they have a weak wind in summer and a mild windy winter. Compared with dry NADORS, the annual averaged relative humidity values in Shuanghu, QOMS, and NAMORS were 40%, 43%, and 52%, respectively. For downward shortwave radiation ($R_{s\downarrow}$) in Figure 2e, Shuanghu station shows the smallest cumulative values, and they were approximately 83, 155, and 218 W m$^{-2}$ higher from

January to the ice-off season in May and 262, 289, and 411 W m$^{-2}$ higher for a complete year in NADORS, NAMORS, and QOMS, respectively. Similarly, the multiyear downward longwave radiation ($R_{l\downarrow}$) has quite similar seasonal variations (Figure 2f), with the highest values appearing in NADORS. Considering Shuanghu station as a reference, the cumulative $R_{l\downarrow}$ in NAMORS, QOMS, and NADORS were 16, 47, and 211 W m$^{-2}$ higher, respectively, for a complete year. Thus, relative to the sum of $R_{l\downarrow}$ and $R_{s\downarrow}$ in Shuanghu, the annual values were 474, 305, and 458 W m$^{-2}$ higher in NADORS, NAMORS, and QOMS, respectively.

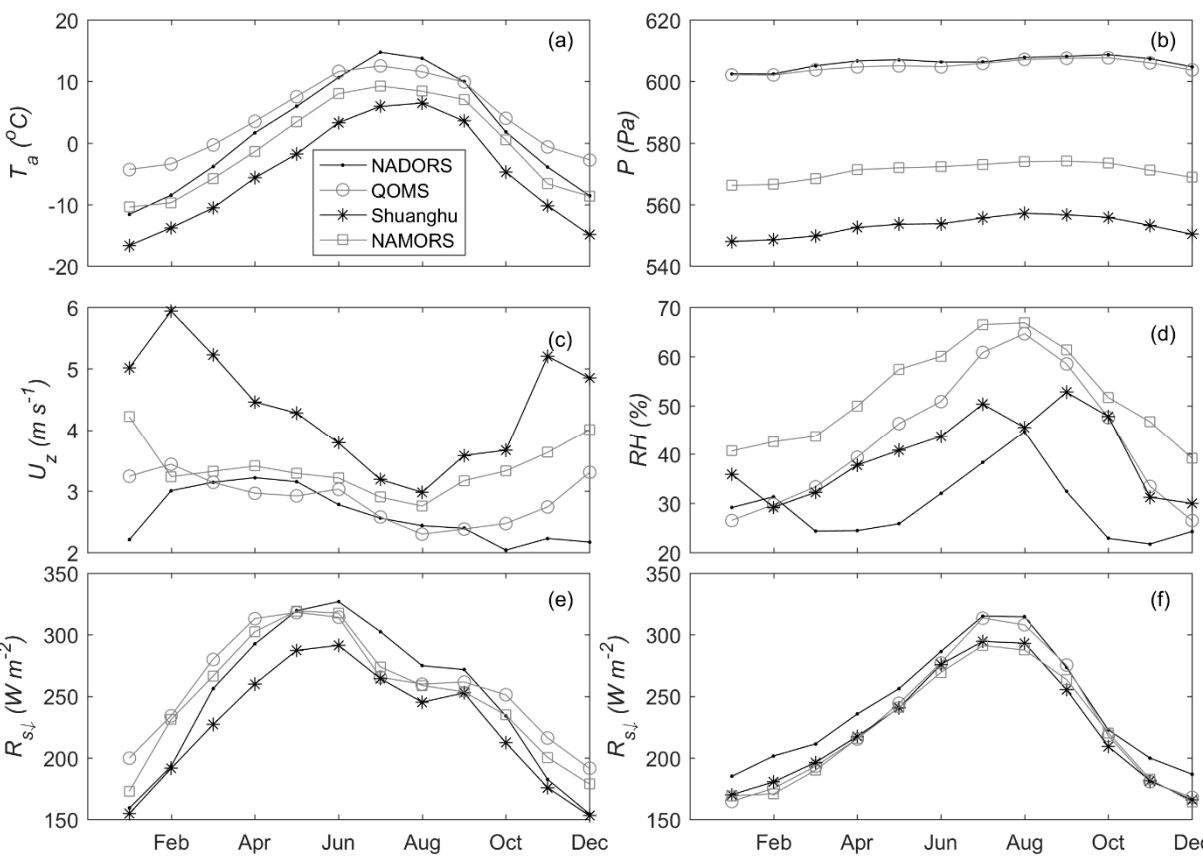

**Figure 2.** The seasonal variations in (**a**) air temperature ($T_a$); (**b**) air pressure ($P$); (**c**) wind speed ($U_z$); (**d**) relative humidity ($RH$); (**e**) downward shortwave radiation ($R_{s\downarrow}$); (**f**) downward longwave radiation ($R_{l\downarrow}$) for NAMORS, NADORS, Shuanghu, and QOMS.

## 2.2. The Stratification and Mixing Dynamics

The stratification onset of a lake can be determined by gradients of temperature, density, and turbulence, which are the commonly used parameters for epilimnion depth ($E_p$) estimation [6,7]. Used as a climate change indicator and reflecting lakes' response to climate change, $E_p$ by different methods are significantly dependent on the threshold values, the vertical data resolutions, and the thermal structure of the water column [1]. In this study, $E_p$ is estimated via the methods of LakeAnalyzer [7] and the "absolute density difference from the surface", where the latter one defined by the shallowest depth with its density 0.1 kg m$^{-3}$ higher than the topmost density has been suggested to be a generic method [1]. Water density is calculated according to the combined contributions of water temperature and salinity. If the density difference between the top and bottom measurements has a value smaller than 0.1 kg m$^{-3}$, the water column is considered to be fully mixed, and $E_p$ equals the lake depth. The salinity profile will also have a significant influence on the water column density distribution and then impact the $E_p$ estimation. Based on the limited density profile measurements in each lake, the influence of salinity gradient is only significant in Daze Co.

In addition, to quantify the relative stability of a lake, we also included the buoyancy frequency ($N^2$) and Schmidt stability ($S_t$), where $N^2$ is the most widely used stratification index and $S_t$ represents the potential energy inherent in the stratification of the water column for resistance to mechanical mixing [32]. The equations are listed as follows:

$$N^2 = \frac{g}{\rho}\frac{d\rho}{dz} \tag{1}$$

$$S_t = \frac{g}{A_s}\int_0^{z_d}(z - z_v)\rho_z A_z \partial z \tag{2}$$

where $N$ is the Brunt–Vaisala frequency (stability frequency) and $N^2$ indicates how much energy is needed for vertical water parcel exchange. $g$ is gravitational acceleration; $\rho$ is the average water density of the water column; $d\rho$ and $dz$ are the differences in water density and depth between the top layer and the bottom layer, respectively. Here, the top layer and bottom layer are defined as 6 m and 34 m, respectively, for a reasonable comparison among the four case study large lakes, while they are 2 m and 13 m for the "small lake". $A_s$ and $A_z$ are lake areas at the surface and at depth $z$, respectively; $z_d$ is the maximum depth of the lake; $z_v$ is the depth to the center of volume of the lake, expressed as $z_v = \int_0^{z_d}zA_z\partial z / \int_0^{z_d}A_z\partial z$. $\rho_z$ is the water density at depth $z$. More details for information on $E_p$ estimation and lake stability parameters can refer to [1,7].

To understand the driving forces of the seasonal variation in $E_p$, the seasonal variations in energy factors (including downward short wave radiation ($R_{s\downarrow}$), downward longwave radiation ($R_{l\downarrow}$), upward longwave radiation ($R_{l\uparrow}$), or water surface temperature ($T_s$), sensible heat flux ($SH$), latent heat flux ($LE$), and heat storage in the water ($R_n - SH - LE$)) and the dynamic forcing factor of $U_z$ were estimated based on meteorological variables close to the lake surface in Nam Co and "small lake". $R_n$ is the net radiation at the water surface. More information about the above-given variables can be found in [24,31].

Moreover, the water heat flux ($G$, W m$^{-2}$) and cumulative heat storage ($G_c$, J m$^{-2}$) of the water can also be estimated by temperature chains and bathymetry information through the following equations:

$$G = \sum_{i=1}^{i=n}\rho c_w d_i \frac{S_i}{S_s}\frac{dT_i}{dt} \tag{3}$$

$$G_c = \sum_{i=1}^{i=n}\rho c_w d_i \frac{S_i}{S_s}\Delta T_i \tag{4}$$

where $c_w$ is the specific heat capacity of water, $d_i$ is the representative depth at layer $i$, $S_i$ is the representative area at layer $i$, $S_s$ is the lake surface area, $T_i$ is the water temperature at layer $i$, $dt$ is the time interval, $dT_i$ is the temperature difference of layer $i$ during $dt$, and $\Delta T_i$ is the temperature difference between $T_i$ and the minimum water temperature.

## 3. Results

### 3.1. The Evolution of Lake Temperature and Heat Storage

The evolution of the average temperature ($\overline{T_m}$) of the water column above 35 m in these high-elevation large lakes shows quite similar seasonal variations, with observed minimum temperatures from December to January and maximum temperatures from August to September (Figure 3). The lowest $\overline{T_m}$ has values of approximately 0.5 °C in the middle of January (23 January 2012, 13 January 2013, and 16 January 2014) in Nam Co, while they are 1.2 °C on 22 December in Bangong Co and 0.3 °C on 13 December in Dagze Co, respectively. For Paiku Co, which was not frozen during winter 2016, the smallest $\overline{T_m}$ with a value of 1 °C appears much later on 12 March. Influenced by water salinity in these four lakes, the lowest $\overline{T_m}$ are all lower than the maximum density temperature (4 °C) of fresh water. In contrast, the highest $\overline{T_m}$ appeared on 17 September 2012 and 6 September 2013 in Lake Nam Co, 30 August in Lake Bangong Co, and 26 August in Lake Dagze Co

and Paiku Co. Compared with the aforementioned four large lakes, the highest $\overline{T_m}$ (average from 0.5 m to 14 m) appeared earlier on August due to the relatively smaller heat capacity of the "small lake".

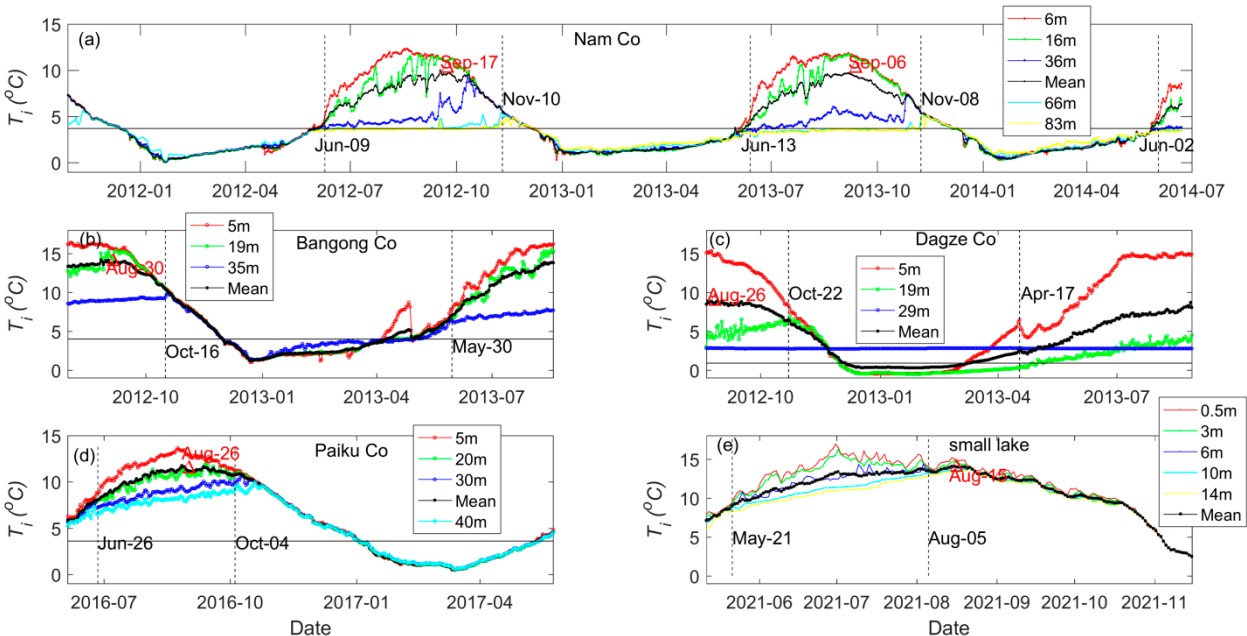

**Figure 3.** The seasonal variations in water temperature at multiple layers ($T_i$) and average water temperature ($\overline{T_m}$) above 35 m at (**a**) Nam Co; (**b**) Bangong Co; (**c**) Dagze Co; (**d**) Paiku Co; (**e**) "small lake". The symbol "mean" indicates the average water temperature at all layers in the "small lake" and the average water temperature appearing in the figure above 35 m; the vertical dotted lines indicate that the temperature gradients between the surface layer and deep layer are approximately 1 °C. The horizontal vertical lines indicate the water temperature at maximum water density.

Similarly, the highest water temperature ($T_{5m}$) at 5 m depth appears relatively earlier than that of $\overline{T_m}$. The highest $T_{5m}$ have values of 16.3 °C in Bangong Co, 15.3 °C in Daze Co, 13.7 °C in Paiku Co, and 12.7 °C in Nam Co. The relative order of $T_{5m}$ is most likely related to lake elevations and depths, where Bangong Co having the highest $T_{5m}$ corresponds to its lowest elevation and highest air temperature in four large lakes. In turn, the water temperature ($T_d$) at a depth of 35 m shows a late peak relative to that of $\overline{T_m}$. The highest $T_d$ has a value of approximately 9.6 °C on 10 October 2016, in Nam Co, and similarly, the values are 10.7 °C and 12.1 °C at a depth of 30 m in Paiku Co and Bangong Co on 9 October 2016, and 4 October 2012, respectively. Because of the high salinity, $T_d$ at a depth of 30 m shows a constant value of approximately 2.6 °C for the whole year, and the water temperature at a depth of 19 m can reach approximately 7 °C at the end of October for Dagze Co. Relative to $\overline{T_m}$, the observed evidence of the relatively advanced peak value in $T_{5m}$ and delayed peak value in $T_d$ indicated the heat transfer process from the surface layer to the deep layer, the change in lake stability, and the seasonal evolution of stratification and mixing during the open water seasons.

After that, the water heat flux ($G$) and cumulative heat storage ($G_c$) in the water of the five lakes were estimated via lake temperature chains and bathymetry data (Figure 4). The $G_c$ values of these lakes have similar seasonal patterns as those of $\overline{T_m}$. Considering the same depth value of 35 m in the four large lakes, the $G_c$ values in Bangong Co, Paiku Co, Nam Co, and Dagze Co have the highest values of $1.37 \times 10^4$ J m$^{-2}$, $1.64 \times 10^4$ J m$^{-2}$, $1.51 \times 10^4$ J m$^{-2}$, and $1.00 \times 10^4$ J m$^{-2}$, respectively, which correspond to the $\overline{T_m}$ values of 14.1 °C, 11.7 °C, 9.7 °C, and 8.9 °C of the four lakes. The much higher $G_c$ values in Paiku Co and Nam Co than in Bangong Co result from the influences of lake bathymetry, where the lake area decreases more slowly with depth in the former two lakes. Furthermore, $G_c$

will be much higher when heat storage in deep layers is involved. For example, the $G_c$ value can amount to $2.08 \times 10^4$ W m$^{-2}$ considering a lake depth of 83 m in Nam Co, where the heat storage in the water below 35 m can account for 25% and 45% of the heat storage from the surface to 83 m during open-water and ice-covered seasons, respectively.

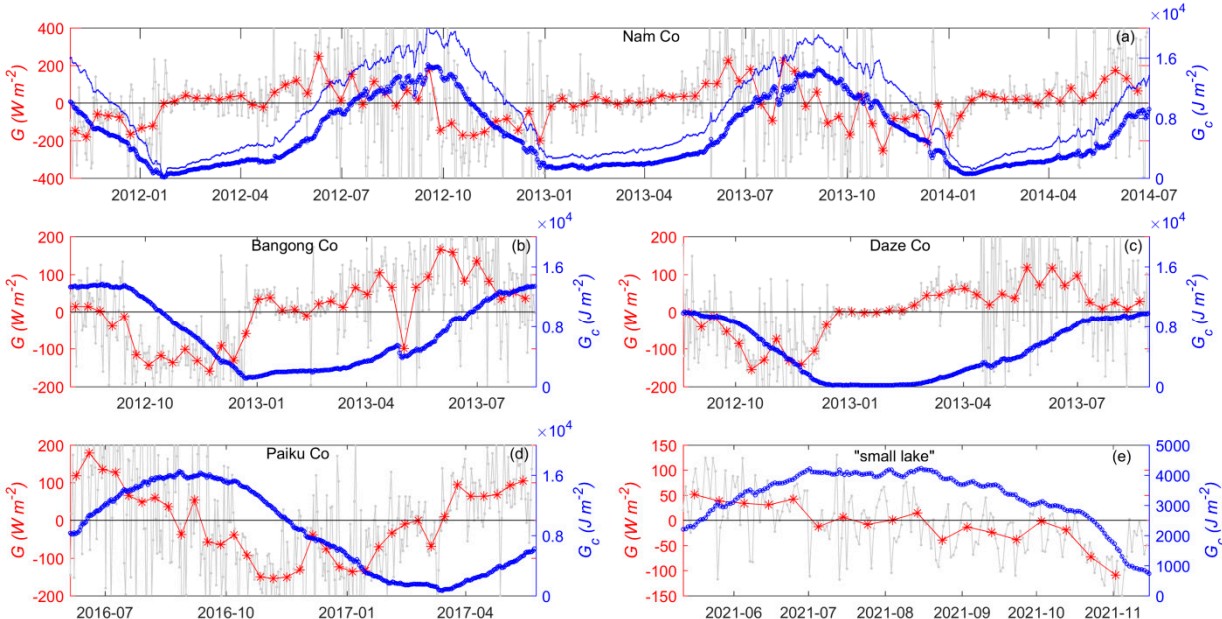

**Figure 4.** The seasonal variations in water heat flux ($G$) and cumulative heat storage ($G_c$) in the lake of (**a**) Nam Co; (**b**) Bangong Co; (**c**) Daze Co; (**d**) Paiku Co; (**e**) "small lake". The "grey dot line" indicates daily heat storage, the "red star line" indicates the 10-day average of heat storage, the "blue star line" represents the cumulative heat storage above 35 m, and the "blue dot line" in (**d**) represents the cumulative heat storage over the whole water column.

Additionally, the heat accumulation periods mainly appeared in May–June, with positive heat storage values ($G$ at a temporal resolution of 10 days) of approximately 115.3 W m$^{-2}$, 87.5 W m$^{-2}$, 145.6 W m$^{-2}$, and 112.4 W m$^{-2}$, respectively, while the heat releasing periods mainly occur in October–November with negative $G$ values of approximately −145.3 W m$^{-2}$, −126.8 W m$^{-2}$, −148.5 W m$^{-2}$, and −125.7 W m$^{-2}$ in Bangong Co, Dagze Co, Paiku Co, and Nam Co, respectively (Figure 4). $G$ values were also quite close to zero during the transition season when $G_c$ values reach their maximum or minimum values. For example, the transition periods were August to September in Bangong Co, July to August in Dagze Co, and September in Paiku Co. These observations explain why the energy budget closure is lower than one during the heat storage period, larger than one during heat release period, and much close to one during the transition period. Compared with the four large lakes, the $G_c$ value in the "small lake" had the largest value of approximately $4.2 \times 10^3$ W m$^{-2}$, which was only half to a quarter of the values in the large lakes. Additionally, the $G$ values at a temporal resolution of 10 days had the highest value of approximately 50 W m$^{-2}$ in May and the lowest value of −108 W m$^{-2}$ at the end of October, where the seasonal variation in $G$ was much smaller than those in large lakes, which is due to its smaller capacity.

### 3.2. The Evolution of Lake Stability and Epilimnion Depth

The lake stability expressed by buoyancy frequency and Schmidt stability shows quite similar seasonal variations, with higher values during the stratification season and close to zero values during the mixing periods (Figure 5). Determined by the density difference between the surface layer (6 m) and the deep layer (34 m), the buoyancy frequency in the four large lakes showed the highest value of $5.0 \times 10^{-4}$ s$^{-2}$ in Dagze Co, followed by

values of $3.6 \times 10^{-4}$ s$^{-2}$ in Bangong Co, $2.0 \times 10^{-4}$ s$^{-2}$ in Nam Co, and $1.9 \times 10^{-4}$ s$^{-2}$ in Paiku Co. The relatively higher buoyancy frequency in Dagze Co resulted from its high-salinity-influenced water density in the hypolimnion. Correspondingly, the largest Schmidt stability values in the four large lakes were 1999, 612, 338, and 670 J m$^{-2}$ in Nam Co, Dagze Co, Bangong Co, and Paiku Co, respectively. Lake depth, salinity, and bathymetry are important factors that can influence Schmidt stability. For example, a lake with a deeper depth, a higher salinity difference, and a cylinder shape would always have a higher Schmidt stability value compared to that with a shallow depth, a low salinity difference, and a circular cone shape. In other words, a lake with the former characteristics always needs more mechanical or thermal energy to reverse its stratification conditions. Similarly, a higher salinity difference between the surface layer and deep layer indicates a strongly stratified condition, which makes it difficult to produce mechanical mixing events. Compared with the four large lakes, the Schmidt stability in the "small lake" is one magnitude lower, with the highest value of approximately 60 J m$^{-2}$. Moreover, the buoyancy frequency (estimated at depths of 2 m and 13 m) has the highest value of $6.6 \times 10^{-4}$ s$^{-2}$, which is higher than those in four large lakes. It indicates that even though stratification can be easily formed in a "small lake", the stratification can also be easily destroyed by mechanical mixing.

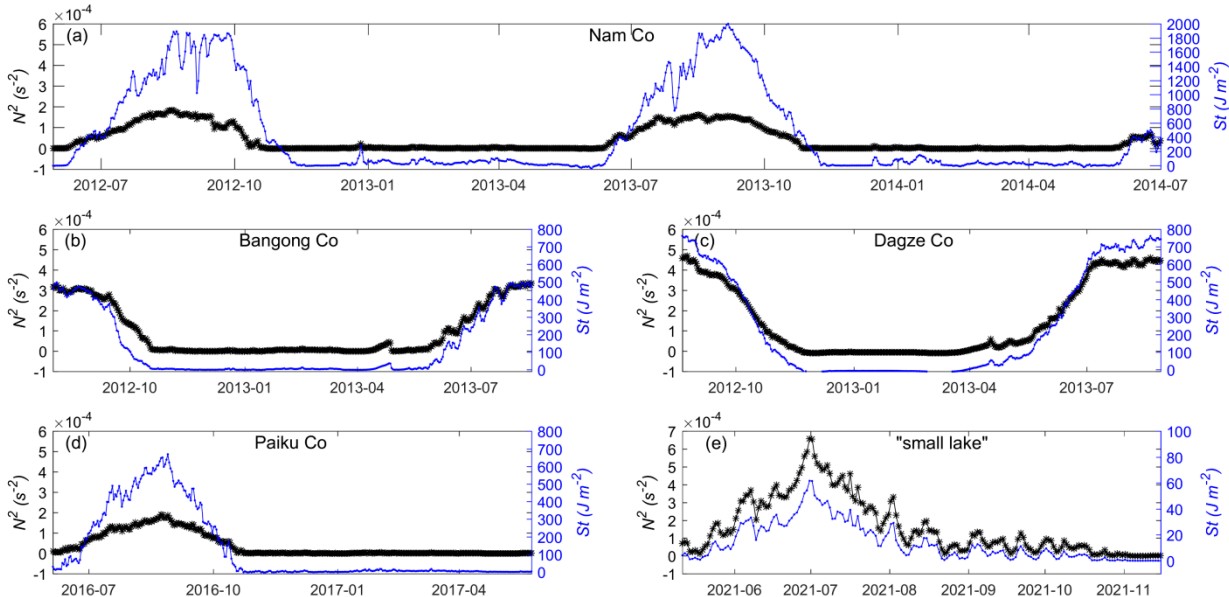

**Figure 5.** The seasonal variations of buoyancy frequency ($N^2$, black line) and Schmidt stability ($S_t$, blue line) in the lakes of Nam Co (**a**); Bangong Co (**b**); Dagze Co (**c**); Paiku Co (**d**); "small lake" (**e**).

The epilimnion depth ($E_p$) estimated by the two methods shows quite similar seasonal variations, with an obvious deepening trend observed during the stratification seasons (Figure 6). All four large lakes can form stratification around May–June and are generally destratified during October–November (Figure 6). In Nam Co, the stratification period lasted for almost 4.5 months, with $E_p$ values of approximately 10–20 m at the beginning (middle of June) and fully mixing to the lake bottom at the end of October (Figure 6a). Similarly, in Bangong Co, the stratification season forms half a month earlier and lasts for approximately 4.5 months from the beginning of June (with $E_p$ values of approximately 10–20 m) to the middle of October (full mixing) (Figure 6b). Lake stratification in Dagze Co also forms in early June and ends in the middle of October, with an obvious increasing trend during its evolution. However, in Paiku Co, the stratification period is one month shorter compared with the other three large lakes, extending from the end of June to the middle of October, during which $E_p$ values show fluctuations rising from approximately 10 m to 40 m. For the "small lake", the stratification forms on 21 May and maintains $E_p$ values

of approximately 3–9 m until 6 August. After that, stratification and full-column mixing events develop alternately until 3 October, with a total of 33 days and 26 days in the lengths of stratification and full-column mixing, respectively. Thus, the total length of stratification in the "small lake" has a value of 112 days during its open water season (10 May to 16 November). In brief, the $E_p$ shows clearly deepening trends during stratification periods in the four large lakes, while the variations in stratification and turnover in the "small lake" are more drastically influenced.

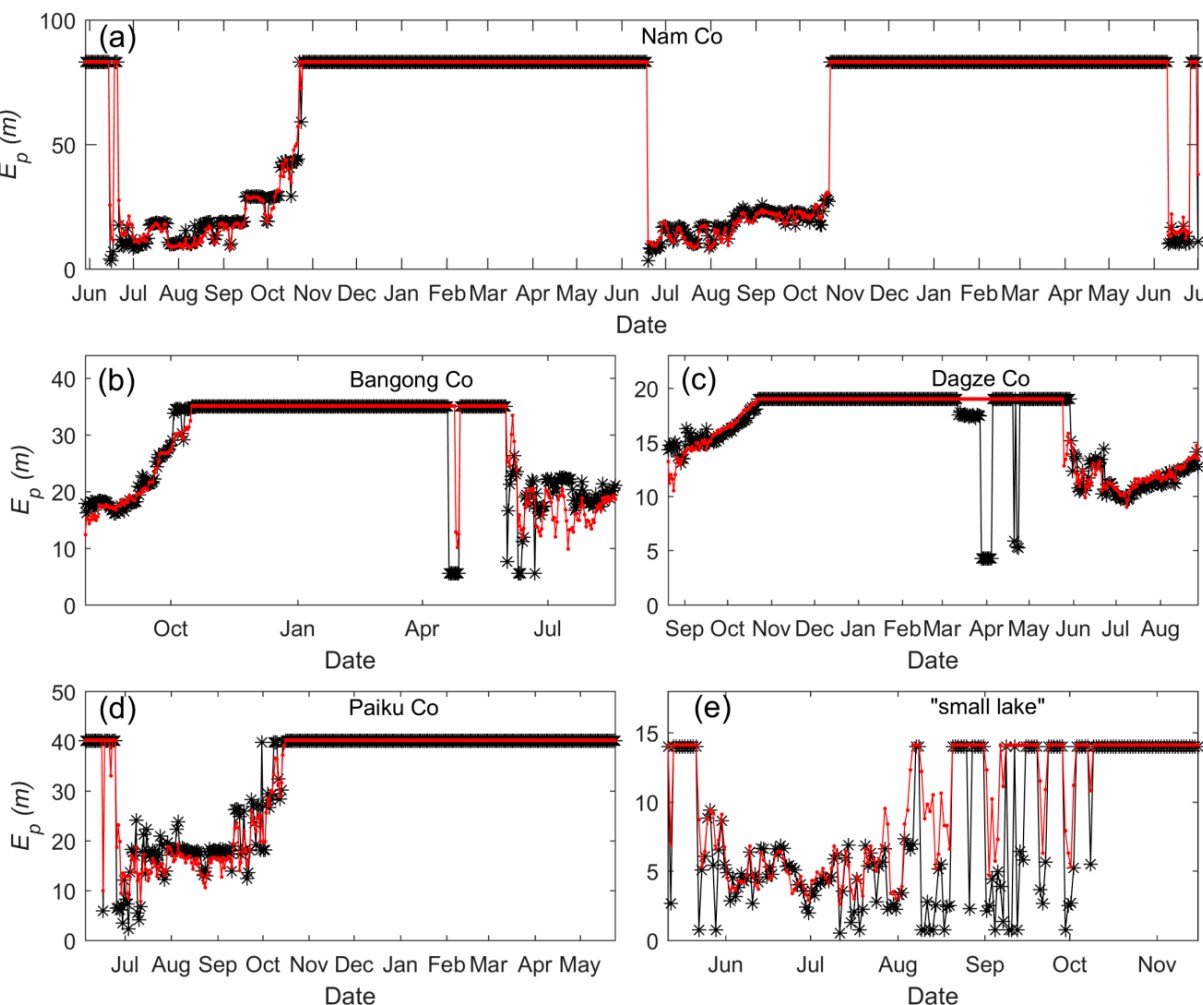

**Figure 6.** The seasonal variations in epilimnion depth ($E_p$) in the lakes of (**a**) Nam Co, (**b**) Bangong Co, (**c**) Dagze Co, (**d**) Paiku Co, and (**e**) "small lake" by the methods of LakeAnalyzer (black star line) and "absolute difference from the surface" (red dot line).

The diurnal variations in $E_p$ cannot be found in Bangong Co and Dagze Co (Figure 7a), where the first layers of water temperature are measured at depths of 5 m and 4 m, respectively. However, $E_p$ shows clear diurnal variations in Nam Co (Figure 7b) and Paiku Co (Figure 7c), where the first layer measurements of the two lakes were 0.4 m and 0.5 m, respectively. Furthermore, when the first layer measurements (0.4 m and 0.5 m) were removed in the calculation of $E_p$ in Paiku Co and Nam Co, the diurnal variations in $E_p$ in the two lakes become weak or almost disappeared. In the "small lake" (Figure 7d), the diurnal variation in $E_p$ was quite obvious, with average $E_p$ values of approximately 6 m at 7:00 and 4 m at 17:00. Similarly, when $E_p$ was estimated from a depth of 3 m, the diurnal variation in $E_p$ nearly vanishes. In summary, the near-surface water temperature is important to investigate the diurnal change in $E_p$ and the deepest $E_p$ occurs at around 17:00.

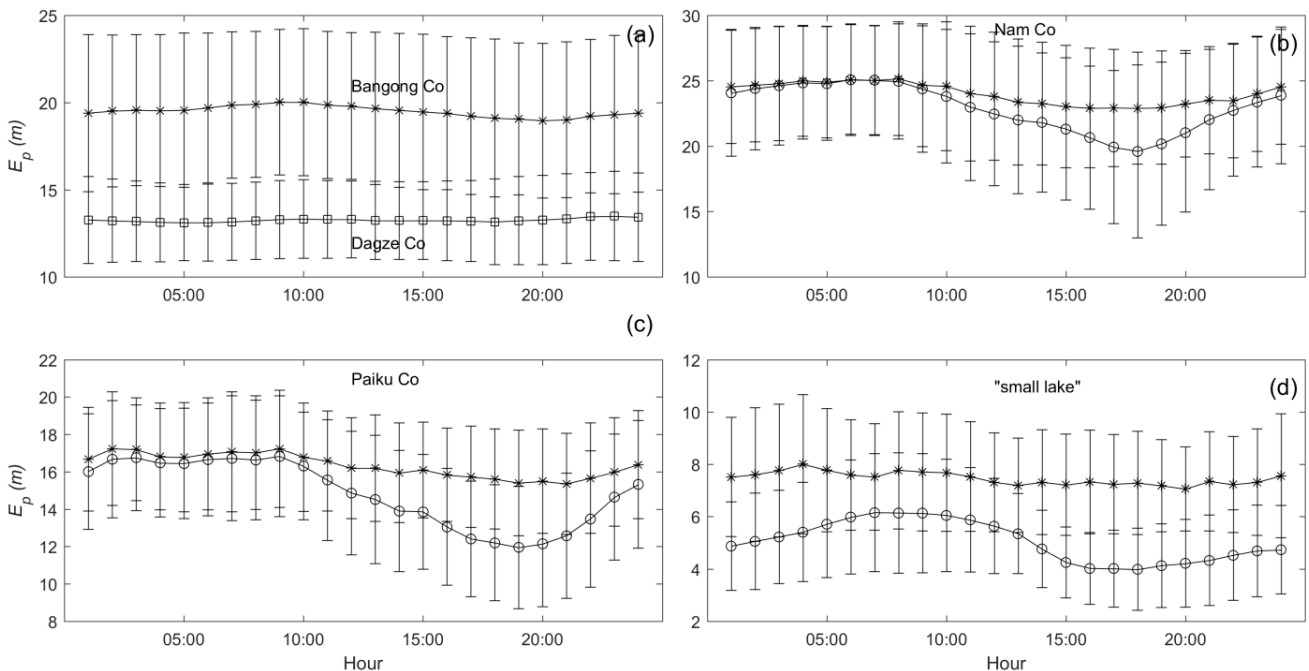

**Figure 7.** The diurnal variations in epilimnion depth ($E_p$) in the lakes of (**a**) Bangong Co (black star line) and Daze Co (black square line); (**b**) Nam Co; (**c**) Paiku Co; (**d**) "small lake". The star line and the circle line indicate the diurnal variation in $E_p$ for temperature chains with and without the layers above 3 m, respectively, in subplots (**b**–**d**).

### 3.3. The Influencing Factors of Epilimnion Depths Variation

The correlation coefficients between $E_p$ and its related influencing factors during the stratification periods of "small lake" (Figure 8) and Nam Co (Figure 9) were estimated as shown in Table 2. Generally, the seasonal variation in $E_p$ is quite similar to that of $U_z$ during the stratification period. In contrast, high radiation heating always corresponded to smaller $E_p$ values. For the "small lake", the largest positive influencing factor was wind force, which shows a correlation coefficient of 0.46 during its stratification period (Figure 8b). The upward longwave radiation ($R_{l\uparrow}$ or $T_s$) shows the largest negative correlation with $E_p$, with a high negative value of −0.59 (Figure 8a). Additionally, $R_{s\downarrow}$ (Figure 8c) and $R_{l\downarrow}$ (Figure 8d) had negative correlation coefficient values of −0.43 and −0.45, respectively, while lake atmosphere interaction turbulent fluxes ($LE$ and $SH$) showed close to zero correlation coefficients in the "small lake". This indicated that both wind-driven mixing and buoyancy-influenced stratification play dominant roles in the mixing and stratification events of the "small lake".

**Table 2.** The correlation coefficients ($r$) between meteorological variables and epilimnion depth.

| $r$ | Year | LE | SH | $R_{l\downarrow}$ | $R_{l\uparrow}$ ($T_s$) | $R_{s\downarrow}$ | $R_n-SH-LE$ | $U_z$ |
|---|---|---|---|---|---|---|---|---|
| Nam Co | 2015 | 0.41 | 0.44 | −0.45 | −0.38 | −0.26 | −0.52 | 0.23 |
|  | 2016 | 0.57 | 0.67 | −0.46 | −0.50 | −0.51 | −0.74 | 0.23 |
| "small lake" | 2021 | 0.04 | −0.02 | −0.45 | −0.59 | −0.43 | −0.35 | 0.46 |

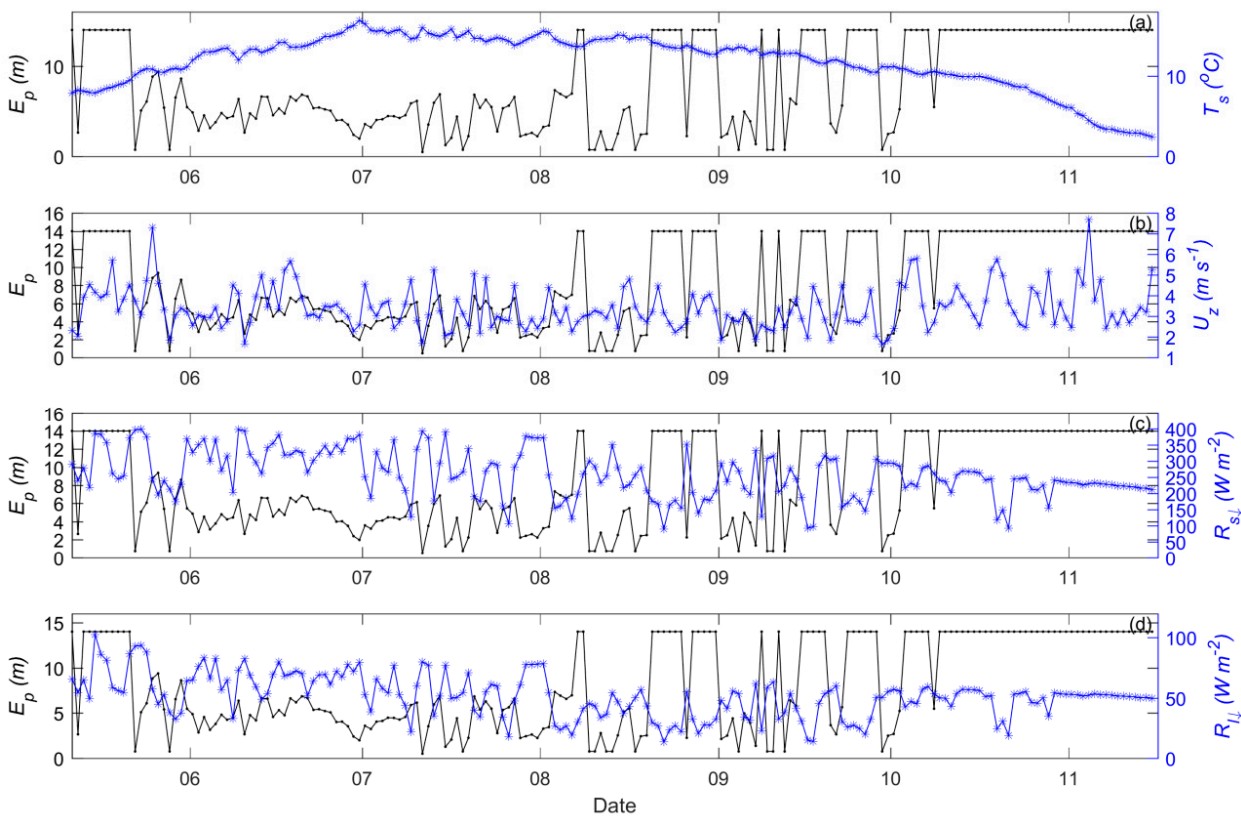

**Figure 8.** The seasonal variations of $T_s$ (**a**); $U_z$ (**b**); $R_{s\downarrow}$ (**c**); $R_{l\downarrow}$ (**d**); during May to November 2021 in the "small lake" in blue colors, where the epilimnion depth ($E_p$) is added in each subplot in black colors.

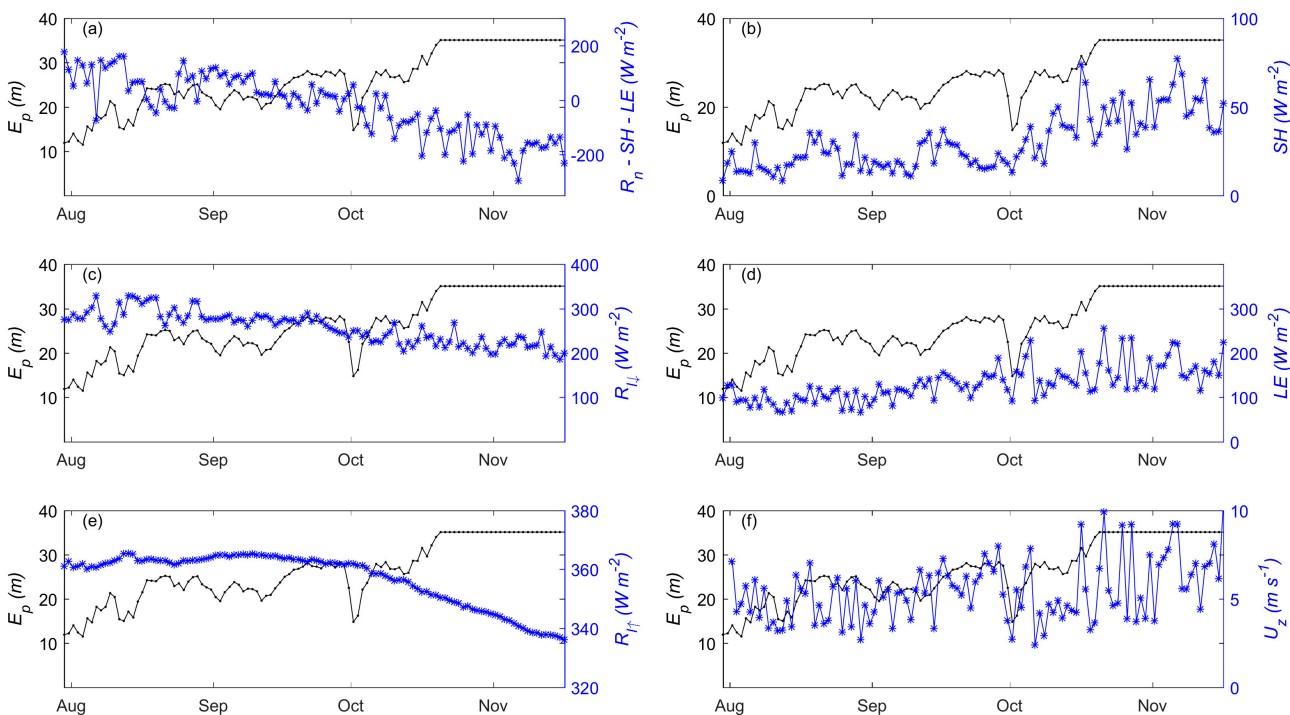

**Figure 9.** The seasonal variations of $R_n - SH - LE$ (**a**); sensible heat flux (*SH*) (**b**); $R_{l\downarrow}$ (**c**); latent heat flux (*LE*) (**d**); $R_{l\uparrow}$ (**e**); $U_z$ (**f**) during the open water season of July to November in Nam Co in blue colors, where the epilimnion depth ($E_p$) is added in each subplot in black colors.

For Nam Co (Figure 9), the factors with positive correlations are *LE*, *SH*, and $U_z$, which had correlation coefficient values of 0.41, 0.44, and 0.23 in 2015 and 0.57, 0.67, and 0.23 in 2016, respectively. The negative influencing factors included $R_{l\downarrow}$, $R_{s\downarrow}$, $R_{l\uparrow}$, and $R_n - SH - LE$, with correlation coefficient values of −0.45, −0.26, −0.38, and −0.52 in 2015 and −0.46, −0.51, −0.50, and −0.74 in 2016, respectively. These results indicated that the heat budget components in Nam Co showed a dominant role in the seasonal variation of $E_p$, where lake–air turbulent heat flux shows obvious positive correlations and $R_{l\downarrow}$ ($T_a$) and $R_{l\uparrow}$ ($T_s$) had obvious negative correlations. However, $U_z$ had much smaller correlation coefficients with the variation of $E_p$ in Nam Co.

## 4. Discussion

The measurements of temperature chains have been established more frequently over high-elevation lakes on the TP, and their seasonal variations may also help to recognize ice phenology events. The prevalence of rapid water warming at the end of ice-melt seasons has been reported recently [25,33], and such a phenomenon can be used to identify the exact ice-melt dates. Furthermore, obvious cooling events with characteristics of a sharp decrease in water temperature and reverse temperature distribution can be identified by their seasonal variations, which may indicate cooling conditions for ice-forming events in shallow waters. Following these characteristics, the ice-forming dates probably appear during the middle of December (19 December 2012, 9 December 2013, and 12 December 2014), and the ice-melt dates occurred at the end of May (26 May 2012, 31 May 2013, and 26 May 2014) for Nam Co (Figure 3a). The dates of ice-forming and ice-melt start relatively earlier in Bangong Co and Dagze Co, with determined dates of 1 December and 29 April in the former and 15 November and 23 April in the latter (Figure 3b,c). Paiku Co shows ice-free status during winter in 2016, and the possible ice-forming and ice-melt dates are 31 December and 15 May, which are estimated by an ice-freezing temperature value of approximately 4 °C (Figure 3d). These determined dates of ice phenology events are quite close to the multiyear average estimations via MODIS products during 2003–2016 [19]. Furthermore, during the winter ice-covered seasons and with water temperatures smaller than $T_{\rho max}$ (the temperature at maximum water density), a reverse temperature distribution of higher water temperature at deep layers and lower water temperature at surface layers appears, which corresponds to a weak stability of the water column during its ice-covered season. Compared with the other three ice-covered large lakes, the smallest $\overline{T_m}$ appears much later in Paiku Co, and the different pattern of $\overline{T_m}$ in the winter season may indicate the effect of the ice freezing process. In ice-covered lakes, the smallest $\overline{T_m}$ most likely corresponds to ice phenology events of fully ice-covered events, at which point the heat storage in the water arrives at its smallest value, then shows a nearly constant value (as in Daze Co) or slowly increasing trends (as in Nam Co and Bangong Co); however, for Paiku Co, which is not frozen in winter [22], the heat releasing period is much longer until March.

Lake surface temperature has been used as the most important variable for lake–atmosphere interaction processes. In the "small lake", the maximum temperature gradient between depths of 0.5 m and 4 m can be as high as 2.5 °C, with average temperature gradient values of 0.9 °C during the stratification period and 0.1 °C during the mixing period. As during 2016 and 2017 in Nam Co, the average temperature gradients between layers of 0.5 m and 3 m have values of only −0.11 °C and −0.17 °C for the stratification period and −0.05 °C and −0.01 °C for the mixing period, with a maximum temperature gradient value of −0.34 °C and −0.85 °C. Similarly, in Paiku Co, the values are −0.15 °C for the stratification period and −0.05 °C for the mixing period, with a maximum temperature gradient value of −0.59 °C. Therefore, it is reasonable to infer that surface warming and cooling may not induce large errors for turbulent heat flux simulation at temporal resolutions of daily and longer. However, the diurnal variation in the simulated turbulent heat flux will definitely be underestimated when temperature measurements at deeper depths are used as a substitution.

Similar to Wilson et al. [1], the largest deviation of the estimated $E_p$ by the two methods mainly occurred for temperature chains with an intensely stratified structure, while the profiles with a three-layered water column structure show quite similar variations. The seasonal variation of $E_p$ by the method of absolute density difference of 0.1 kg m$^{-3}$ from the surface water is more stable and was suggested for wide application. Furthermore, the water temperatures above the estimated $E_p$ show similar seasonal variations, with much larger amplitudes in shallow layers and much smaller amplitudes in deep layers. This may represent the common heat transfer phenomenon in which much stronger surface warming and cooling appear for surface waters. For example, in Dagze Co, the amplitudes of the water temperature above $E_p$ gradually decrease from 4 m to 13 m. Because of the high elevation and strong solar heating on the TP, the surface warming during the day, and surface cooling at night should indicate a strong diurnal variation in stratification and mixing in these high-elevation lakes. During the stratification period, surface warming can heat the surface layer during the day and form a stable water column, which prohibits mixing events. At night, strong surface cooling can reduce the surface water temperature, enlarge the water density, and promote a downward mixing event. Thus, the high amplitude of $T_s$ in surface layers should correspond to the observed diurnal variation of $E_p$. For example, in the "small lake", the amplitude of $T_i$ has obvious diurnal variations to a depth of 3 m, which corresponds to the diurnal variation of $E_p$, with an average $E_p$ of approximately 5 m. In contrast, when the measurements from 4 m to 14 m were used, the average $E_p$ will deepen to 7 m, with no obvious diurnal variation appearing (Figure 7d). Therefore, it may be justified that diurnal variations in $E_p$ may be widespread and that surface water temperature measurements at layers shallower than 3 m are important for identifying such phenomena. Moreover, the simulated lake surface temperature by the Flake model has been evaluated primarily at daily resolution [34,35]; however, the diurnal variation of the simulated lake surface temperature, which should be related to the diurnal variation of the epilimnion depth, should be paid more attention.

The thermal stratification and mixing dynamics in all five studied lakes show dimictic patterns, where continuous stratification periods can be observed. Recently, a special summer destratification phenomenon over a large and deep high-elevation lake (Langa Co, with an area of 258.9 km$^2$ and a maximum depth of 40 m) has been reported [36], where a discontinuous polymictic pattern contradicts the typical dimictic pattern of these high-elevation large and deep lakes. The lake bathymetry (with a shallow depth in the northern basin and a deep depth in the southern basin) and the strong combined glacier and mountain valley winds may induce basin-scale circulations, which may be the reasons for destroying the lake stratification in summer seasons. However, based on the published temperature measurements in these high-elevation lakes, the majority show dimictic characteristics for thermal stratification and mixing conditions.

## 5. Conclusions

The evolution of lake thermal stratification and mixing is an important process for studying lake heat and gas transport in the water column and shows significance for lake–atmosphere interactions and modeling lake regional climate effects. A comparative study of the seasonal variations in epilimnion depth and stability parameters via water temperature chains and meteorological variables over five high-elevation lakes (Nam Co, Bangong Co, Daze Co, and Paiku Co, "small lake" adjacent to Nam Co) indicates the following:

(1) The epilimnion depths in these high-elevation large lakes show a dimictic pattern at their seasonal variations, forming at a depth of approximately 10–15 m after spring turnover and evolving with a gradually increasing trend until autumn turnover. The diurnal variation in epilimnion depth can be evidenced by water temperature chains with layers shallower than 3 m depth, i.e., in Nam Co, Peiku Co, and "small lake".

(2) The seasonal variation in the epilimnion depth of the "small lake" keeps stratification during the whole heat storage period and shows stratification and mixing events alternatively during the heat release period.

(3) For Nam Co, the radiation and heat budget components show a dominant role in the seasonal variation in epilimnion depth, with much smaller influences by the wind. In contrast, the radiation budgets and heat storage in the water show negative correlations, and wind speed shows a significant positive correlation but with no correlation with sensible heat flux and latent heat flux.

These findings are important for understanding the lake processes of these high-elevation lakes and should provide significant validation datasets for modeling high-elevation lake processes.

**Author Contributions:** Conceptualization, B.W. and Y.M.; Methodology, Y.W. and B.W.; Formal analysis, W.M.; Investigation, L. and L.W.; Data curation, Y.W. and L.W.; Writing—review & editing, Y.W., L., W.M. and B.S.; Supervision, Y.M. and B.S.; Project administration, B.W. and Y.M.; Funding acquisition, B.W. and Y.M. All authors have read and agreed to the published version of the manuscript.

**Funding:** This research was jointly funded by the National Natural Science Foundation of China (Grant Nos. 42075085, 91837208, 41830650), the Second Tibetan Plateau Scientific Expedition and Research Program (Grant No. 2019QZKK0103), the Strategic Priority Research Program of the Chinese Academy of Sciences (Grant No. XDA20060101), and the Youth Innovation Promotion Association of the Chinese Academy of Sciences (2022069).

**Data Availability Statement:** Publicly available datasets were analyzed in this study. This data can be found here: https://data.tpdc.ac.cn/en/.

**Acknowledgments:** We thank Junbo Wang for providing temperature chain measurements in Nam Co, Yanbin Lei for temperature chain measurements in Paiku Co, and Mingda Wang for temperature chain measurements in Daze Co and Bangong Co. These temperature chain measurements and meteorological variables in can NAMORS, QOMS, Shuanghu, and NADORS can be accessed in the National Tibetan Plateau Third Pole Environment Data Center (https://data.tpdc.ac.cn/en/).

**Conflicts of Interest:** The authors declare no conflict of interest.

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
