# Peer review of "Analysis of Lake Stratification and Mixing and Its Influencing Factors over High Elevation Large and Small Lakes on the Tibetan Plateau"

_water, doi:10.3390/w15112094_

Round 1

Reviewer 1 Report

The article deals with one of the basic limnological characteristics of water stratification and mixing. The layout of the article is generally well organized and the content is clear. Noteworthy is the rich collection of measurement data used in the study. I recommend the article for publication after taking into account the following comments.

I do not find a clearly defined purpose of the paper. There are a few research questions posed, but this is still not the goal. In the rest of the paper, it should be shown whether the purpose of the research was fulfilled. If there were any limitations, it should be shown which ones.

How did the sensors function during the winter (especially for the shallowest depths)? In addition, what was the structure of the ice? Radiation penetration and consequently density changes under the ice cover depend on it.

I do not find information on wind direction. With reference to the location of the lake (its orientation), this has significant implications for generating more or less ripple and, consequently, for the depth of mixing.

The discussion chapter lacks a broader reference of the results obtained for the functioning of these ecosystems (hydrobiology, water quality, etc.).

Certain statements in the chapter conclusions are obvious and do not require detailed research:

"The seasonal variation in epilimnion depth can show stratification and mixing events alternatively in the "small lake", which has much lower lake stability and can be easily influenced by driving forces."

The literature record needs to be adjusted to MDPI standards.

Reviewer 2 Report

General comments:

The article collected in situ measurements of lake temperature chains and surrounding meteorological variables over four large lakes and a “small lake” to investigate the seasonal and diurnal variation of epilimnion depth. The study analysed the differences in driving forces and interesting results were reported. For example, the Ep estimated by the two methods showed similarity in results. The stratification and mixing in the four large lakes showed a dimictic pattern, with obvious spring and autumn turnovers. Also, they form stratification during heat storage periods, with E_p quickly locating at depths of approximately 10-15 m, and after that E_p increases gradually to the lake bottom. Additionally, the diurnal variation in E_p can be obtained both in the large and small lakes when temperature measurements above 3 m depth are included.

The manuscript presented is fairly good-organized theoretical work and is an original contribution to the topic. The manuscript would be of interest for the readership of the journal.

However, there are several issues concerning the methods and presentation of the results, which require to be addresses scientifically.

Below are my suggestions

General opinion:

Title: Title is appropriate.

Abstract: Overall, the level of abstract is poorly presented. This section needs to be re-organized by following the journal style of presenting abstract. It suffers from lack of some objectives of this work. Furthermore, abstract must be much more quantitative regarding the results.

Minor comments

Check use of superscript in kgm-3 and subscript in E_p. May using Ep is better. Also, too many uses of quotation marks. Please first use is ok. No need to repeat them.

Keyworks: the provided keywords are appropriate and would improve the article search results in the future or increase the article's visibility to a large audience.

Introduction: Overall level of this section is well presented and relevant literature were highlighted. The literature review is more critical and straight to the point. I like the flow and relevant references are included. However, few issues need to be properly clarified. See minor comments below

Minor comments

Editorial check for proper citation required (see journal style)

Clearly highlights the study’s motivation.

Materials and methods: The methodology was standard and reproducible. This section presentation is fairly described. However, no information on the potential assumption is mentioned. I expected that information about the data quality will be helpful. Particularly, the citation of gauge instrument information and type may helpful. I noticed; the authors used data collected by Ma et al. 2020, Right? Did they perform any quality checks and if so can you provide a sentence on that.

Minor comment:

Line 147-149: please rephrase “The climatic background of these high-elevation lakes can be provided by multiyear average meteorological variables by long-term observations at comprehensive research and observation stations …”

Line 253-282: The equations are not properly presented. Check and correct them

Authors should clarify issues of assumptions made in the study and how these assumptions may impact the results.

Results: This section is clear and presented in a logical sequence. The provided figures and table are clear and necessary for the presentation of the results.

Minor comments

Maintain consistency in use of E_p and Ep

Discussion: I find the discussion to be reasonable and largely supported by the results. This section is well presented and discussed. I like the presentation here. I have minor suggestion to contribute here. The authors can highlight how this study bridges the gap between theory and practice. In addition, the implication of this findings to the managers of the facility will be useful.

Conclusions: this section need to be reorganized. Presenting this section in bullets form maybe helpful to readers.

References: Relevant references are included in the paper.

Minor language editing required.
